# Genome-Wide Identification and Expression Analysis of *Isopentenyl transferase* Family Genes during Development and Resistance to Abiotic Stresses in Tea Plant (*Camellia sinensis*)

**DOI:** 10.3390/plants11172243

**Published:** 2022-08-29

**Authors:** Liping Zhang, Min Li, Jianyu Fu, Xiaoqin Huang, Peng Yan, Shibei Ge, Zhengzhen Li, Peixian Bai, Lan Zhang, Wenyan Han, Xin Li

**Affiliations:** 1Key Laboratory of Tea Quality and Safety Control, Ministry of Agriculture and Rural Affairs, Tea Research Institute, Chinese Academy of Agricultural Sciences, Hangzhou 310008, China; 2College of Horticulture Science and Engineering, Shandong Agricultural University, Tai’an 271018, China; 3State Key Laboratory of Crop Biology, Shandong Agricultural University, Tai’an 271018, China

**Keywords:** abiotic stress, expression pattern, gene family, growth and development, isopentenyl transferase (IPT), tea plant

## Abstract

The tea plant is an important economic crop and is widely cultivated. Isopentenyl transferase (IPT) is the first and rate-limiting enzyme of cytokinin (CK) signaling, which plays key roles in plant development and abiotic stress. However, the *IPT* gene family in tea plants has not been systematically investigated until now. The phylogenetic analyses, gene structures, and conserved domains were predicted here. The results showed that a total of 13 *CsIPT* members were identified from a tea plant genome database and phylogenetically classified into four groups. Furthermore, 10 *CsIPT* members belonged to plant ADP/ATP-*IPT* genes, and 3 *CsIPTs* were tRNA-*IPT* genes. There is a conserved putative ATP/GTP-binding site (P-loop motif) in all the CsIPT sequences. Based on publicly available transcriptome data as well as through RNA-seq and qRT-PCR analysis, the *CsIPT* genes which play key roles in the development of different tissues were identified, respectively. Furthermore, *CsIPT6.2* may be involved in the response to different light treatments. *CsIPT6.4* may play a key role during the dormancy and flush of the lateral buds. *CsIPT5.1* may play important regulatory roles during the development of the lateral bud, leaf, and flower. *CsIPT5.2* and *CsIPT6.2* may both play key roles for increased resistance to cold-stress, whereas *CsIPT3.2* may play a key role in improving resistance to high-temperature stress as well as drought-stress and rewatering. This study could provide a reference for further studies of *CsIPT* family’s functions and could contribute to tea molecular breeding.

## 1. Introduction

As phytohormones, cytokinins (CKs) play essential regulatory roles in many plant developmental processes and abiotic stress responses. For example, previous studies showed that CK signaling is involved in the control of light perception, meristem activity, shoot branching, as well as the development of leaves, flowers, and seeds, etc [1]. Isopentenyladenine (iP)- and *trans*-zeatin (*t*Z)-type CKs are negative regulators of plant adaptation to abiotic stresses [2]. CK levels are directly regulated by CK metabolic genes. *Isopentenyl transferase* (*IPT*) genes exist in plants and bacteria and play key roles in the first step of CK synthesis [2]. In plant, *IPTs* contain adenylate-*IPTs* (ADP/ATP-*IPTs*), Class I and Class II tRNA-*IPTs* [2,3,4]. The ADP/ATP-*IPTs* were only found in angiosperms. The Class I tRNA-*IPTs* may represent direct successors of the *miaA* genes which were found in all plant genomes, whereas Class II tRNA-*IPTs* originated from prokaryotic genes [2]. Previous studies have suggested that ADP/ATP-IPTs controls the biosynthesis of iP- and *t*Z-type CKs, whereas tRNA-IPTs are mainly involved in the biosynthesis of *cis*-zeatin (*c*Z)-type CKs [4,5,6]. To date, 9, 10, and 9 *IPT* genes have been found in *Arabidopsis* [5,7], rice [8], and *Populus* [6], respectively. In *Arabidopsis*, *AtIPT1* and *AtIPT3-AtIPT8* encode ATP/ADP-IPTs, and the other two *AtIPT* genes, *AtIPT2* and *AtIPT9*, encode tRNA-IPTs [5,9].

The roles of *IPT* genes have been extensively studied [1,5,8,10]. The transgenic plants overexpressing *ZmIPT2* and *GmIPT1* both the typical phenotypes of CKs overproduction [10], whereas the ADP/ATP-*IPTs* mutant exhibited reduced inflorescence [5]. Previous studies showed that ADP/ATP-*IPTs* are negative regulators of plant response to abiotic stresses [5,8,10]. For example, *Arabidopsis* quadruple ADP/ATP-*IPT* mutant *ipt1;3;5;7* displayed markedly enhanced tolerance to salt and drought stresses [5]. However, the transgenic peanuts overexpressing *IPT* gene resulted in increased drought tolerance [1]. The improved tolerance to abiotic stresses in plants can be obtained through using a stress/senescence-associated promoter to drive ADP/ATP-*IPT* expression [5].

The tea plant (*Camellia sinensis*) grows widely, and the tea is tremendously popular as a non-alcoholic beverage all over the world [11,12]. The development and the stress resistance of the tea plant determines its yield and quality. Dormancy is a temporary cessation of the growth of the terminal buds. Previous study has suggested that endogenous CKs increase during tea dormancy release [13]. In our recent study, the expression and functional analysis showed that the 3′ UTR alternative splicing (AS) variant 2 of *Cs*ADP/ATP-*IPT5* might act as the predominant AS transcript in the shoot branching processes of tea plants [14]. Tea plants thrive in the wild and is seriously affected by various abiotic stresses, including salt, drought, cold, and heat, etc [11]. Transcriptomic analysis showed that CK biosynthetic genes were up-regulated under drought-stress, but down-regulated during recovery [15]. However, only a little is known about the roles of CK signaling in the regulation of tea plant development and stress responses so far. 

Gene family analysis is a useful approach for a better understanding of the gene structure and function, etc. [4], and a systematic study of the *IPT* gene family in tea plant has not been conducted currently. Given the key roles of *IPT* genes in regulating plant development and abiotic stress, here, a total of 13 *CsIPT* family members were identified based on the genomic databases of the tea plant cultivar (cv.) Longjing 43 (LJ43) and cv. Shuchazao (SCZ). The phylogenetic relationships, conserved domains, and conserved motifs of *CsIPT* genes were systematically analyzed. Furthermore, based on publicly available transcriptome data as well as through RNA-seq and qRT-PCR analysis, the expression patterns of *CsIPT* numbers in different tea tissues as well as during development and abiotic stress processes of tea plants were analyzed. Based on the above results, some candidate *CsIPT* genes which play key regulatory roles in tea plants were preliminarily screened. The results could provide a valuable foundation for further studying the functions of *CsIPT* family numbers in the tea plant.

## 2. Results and Discussion

### 2.1. Identification and Bioinformatics Analysis of Tea Plant CsIPT Genes

Based on the genomic database of two tea plant cultivars, the *CsIPT* family members were identified using all annotated *Arabidopsis AtIPT* genes in the TAIR as query sequences. A total of seven and nine *CsIPT* genes were identified from the genome of LJ43 and SCZ, respectively. After manual filtering and merging, a total of 13 unique Cs*IPT* genes with confirmed conserved domains were identified and annotated.

The CDS length of the identified *CsIPT* genes ranged from 447 to 2286 bp, encoding peptides ranging from 149 to 761 amino acids, where CsIPT6.4 is the smallest and CsIPT9.2 is the largest. The expected molecular weights (MW) ranged from 15.90 to 55.42 kDa. It is worth mentioning that the length and size of CsIPT2, CsIPT9.1, and CsIPT9.2 proteins were all significantly larger as compared with the rest, which is similar to that of IPT2 and IPT9 in *Arabidopsis* and rice [8]. The theoretical isoelectric point (pI) showed a wide range of varying from 5.77 to 8.99. Among them, seven are acidic and six are basic in nature, indicating the nature of CsIPT proteins (Table 1).

Except for CsIPT7.2, the grand average of hydropathy (GRAVY) of CsIPT proteins varied from −0.705 to −0.112, indicating that these 12 CsIPT proteins are hydrophilic in nature, whereas the GRAVY of CsIPT7.2 is 0.088 and is a hydrophobic protein. There are four CsIPTs, including CsIPT2, CsIPT3.2, CsIPT7.1, and CsIPT7.2, whose instability index is below 40, indicating that they are relatively stable. However, the instability index of the other nine CsIPT proteins exceeded 40 (40.54–52.07), indicating that they are unstable (Appendix A). Variable subcellular localization and the presence or absence of glycosylation sites within members of each family predicts their variable functions and substrate specificities [1]. Protein glycosylation contributes to the regulation of enzymatic activity, translocation, and protein stability [10]. The glycosylation sites were predicted to occur in all of the CsIPT proteins. Therefore, the glycosylation sites add the complexity to *CsIPT* regulation. The prediction of subcellular localization showed that CsIPT6.3 may be localized in the chloroplasts. The likelihood of CsIPT9.2 localization in the chloroplasts and mitochondria is 51.1% and 46.15%, respectively. However, the localization of the other 11 CsIPT proteins is not yet understood. None of the CsIPTs had a signal peptide, indicating that CK synthesis that is catalyzed by CsIPTs takes place in specialized tissues or cells.

### 2.2. Multiple Sequence Alignment and Phylogenetic Analysis of CsIPT Gene Family

Phylogenetic analysis based on sequence similarity is a powerful tool to predict orthologous genes and their functions [1]. To identify subgroups and reveal the evolutionary relationships of the tea plant *CsIPT* gene family, multiple sequence alignment of 13 CsIPTs and their *Arabidopsis* homologs was performed, and an NJ phylogenetic tree was generated. The results showed that except for *AtIPT1, AtIPT4*, and *AtIPT8*, most of the identified *CsIPTs* had homologs in *Arabidopisis*, indicating that the expansion of the *IPT* members may have occurred before the divergence of the tea plant and *Arabidopsis*. Naming newly identified genes based on their orthologs in closely related species is a systematic way forward [1]. In this study, a systematic approach was followed and *CsIPT* family members were renamed according to their orthologs in *Arabidopsis*.

According to the phylogenetic analysis, *CsIPTs* and *AtIPTs* were clustered into four subgroups: Group I (*CsIPT6.1*, *−6.2*, *−6.3* and *−6.4*), Group II (*CsIPT2*), Group III (*CsIPT3.1*, *−3.2*, *−5.1*, *5.2*, *−7.1* and *−7.2*), and Group IV (*CsIPT9.1* and *CsIPT9.2*). Group I and Group III included the majority of the *CsIPT* genes, which belonged to plant ADP/ATP-*IPT* genes. On the other hand, the tea plant tRNA-*IPT* genes were highly diverse. The three genes could be divided into two subgroups: *CsIPT2* in Group II, the *AtIPT2* homolog, is a plant tRNA-*IPT* gene of eukaryotic origin; *CsIPT9.1*, *CsIPT9.2*, and *AtIPT9* all shared the properties of plant tRNA-*IPT* genes with prokaryotic origin, and were classified into Group IV. This result is consistent with previous studies, which suggested that the total number of tRNA-*IPT* genes in each flowering plant species is either two or three and is conserved in angiosperm genomes [5].

In most branches, the *CsIPT* genes were closely clustered with orthologs from *Arabidopsis*, and the closely related orthologous genes clustered in similar subgroups that commonly exhibit functional similarities [11]. The results revealed that three of *CsIPT* members (*CsIPT2*, *CsIPT9.1*, and *CsIPT9.1*) were tRNA-*IPT* genes, along with two in *Arabidopsis* (*AtIPT2* and *AtIPT9*) (Figure 1). Previous studies suggested that in angiosperm genomes, the total number of tRNA-*IPTs* is either two or three and is conserved, whereas the numbers of ATP/ADP-*IPTs* exhibit considerable expansion and diversification among species [5]. Thus, the number of ADP/ATP-*IPT* and tRNA-*IPT* genes that were identified in the tea plant is consistent with that in other angiosperms.

Further phylogenetic constructions were performed to confirm the *CsIPT* genes subgroups and investigate their evolution using IPT sequences from various species, including dicotyledons, monocotyledons, moss, algae, and bacteria. The IPT protein sequences of these species are listed in Appendix A. The results showed that the corresponding *IPT* genes in seven plants were divided into four groups: Groups I, II, III, and IV. *CsIPT* genes are asymmetrically distributed in four subgroups. The *IPT* genes in dicotyledons and monocotyledons were randomly dispersed among the four groups. This phenomenon suggested that *IPT* genes that originated from both dicotyledons and monocotyledons were not lost after the divergence of the two plant groups. Furthermore, six *IPT* genes in the bryophyte *P. patens* (*PpIPTs*), eight *IPT* genes in the bacteria (*IPTZ_AGRT5*, *IPTZ_AGRT7*, *MIAA.RHIRD*, *IPT_RHIAD*, *IPT_DICDI*, *MIAA_DICDIb*, *MIAA_DICDIc1*, and *MIAA_DICDIc2*), and three *IPT* genes in algae (*alVctIPT_I*, *alMptIPT_I*, and *alMptIPT_P*) were found in Group IV, which also included the plant tRNA *IPT* genes of prokaryotic origin (*BrIPT9.1, BrIPT9.2*; *AtIPT9*; *SlIPT6*; *CsIPT9.1*, *CsIPT9.2*; *ZmIPT10*; *OsIPT10*; *PtIPT9*), implying that these *IPT* genes in bryophyte, bacteria, and algae possibly originated from the tRNA-*IPT* genes (Appendix A). The presence of evolutionary subgroups suggests the unique functional properties for each subgroup. *AtIPT2* from *Arabidopsis* is actually a tRNA-*IPT* gene and is responsible for the synthesis of cZ-type CKs. Based on the sequence and gene structure similarities, newly identified *CslPT9.1* and *CslPT9.2* in tea plants may have a similar function.

Comparisons of IPT proteins between tea plant and *Arabidopsis* were conducted to characterize the IPT sequences. The results of multiple alignment showed that CsIPT2 and AtIPT2 contained three inserted regions, thus the carboxyl-terminal region of CsIPT2 and AtIPT2 sequences had an extra 102 and 94 amino acids, respectively (the sequences were put in three red boxes). On the other hand, in the carboxyl-terminal region, AtIPT9, CsIPT9.1, and CsIPT9.2 all had four discontinuous inserted regions of 93, 65, and 93 extra amino acids, respectively (the sequences were put in four red boxes). Furthermore, among these inserted regions, two regions were conserved in CsIPT2, CsIPT9.1, and CsIPT9.2. The consensus sequences were classified as their respective tRNA binding sites of the two types of plant tRNA CsIPT proteins. It also can be seen that the two types of plant tRNA-CsIPT proteins had different consensus sequences. Lastly, the multiple alignment of CsIPT with AtIPT protein sequences revealed a conserved region (the sequences were put in black boxes, and the text were in shallow color). It was found in all the CsIPT protein sequences and was identified as a putative ATP/GTP-binding site (P-loop motif), with a core sequence of TGxGKS (Motif 1), which were underlined with a rough black line (Appendix A).

### 2.3. Gene Structure and Conserved Domain Analysis of CsIPT Genes

There were four *Cs*ADP/ATP-*IPTs* (*CsIPT3.2*, *CsIPT6.2*, *CsIPT6.3*, and *CsIPT6.4*) that only had a single exon without introns, the other four *Cs*ADP/ATP-*IPTs* (*CsIPT5.1*, *CsIPT6.1*, *CsIPT7.1*, *CsIPT7.2*) had three exons and two introns, and two *Cs*ADP/ATP-*IPTs* (*CsIPT3.1*, *CsIPT5.2*) were composed of two exons and one intron. On the other hand, three *Cs*tRNA-*IPTs* could be further divided into plant tRNA-*IPT* genes of eukaryotic (*CsIPT2*) and prokaryotic (*CsIPT9.1* and *CsIPT9.2*) origin. However, *CsIPT2* and *CsIPT9.1* both had 11 exons and 10 introns, respectively, whereas *CsIPT9.2* had 18 exons and 17 introns, which was consistent with the more complicated tRNA-*IPT* gene structures in other higher plants (Figure 2a).

The position of the conserved IPT domain on each protein was illustrated to get an insight of the protein structure. A total of 10 CsADP/ATP-IPT protein sequences all possess a common conserved domain P-loop NTPase superfamily, respectively. Furthermore, two CstRNA-IPT (CsIPT9.1 and CsIPT9.2) protein sequences both contain a PLN02840 multi-domain, and CsIPT2 contains a PLN02748 multi-domain, respectively (Figure 2b). The above results showed that all of the CsIPT protein sequences harbored the characteristic conserved domains of the *IPT* gene family. Furthermore, the domain analyses clarify the demarcation between the IPT domain of tRNA-IPT and ATP/ADP-IPT. That is to say, all of the identified CsADP/ATP-IPT proteins here share a conserved putative ATP/GTP binding site (P-loop motif).

There were 10 conserved motifs that were found for the CsIPT protein sequences. All of the CsIPTs showed conserved motif structures among the members of every type, which is consistent with the *IPT* gene family in other plants. Significant differences were observed among the different groups. To be specific, motif 7 could be specifically detected in CsADP/ATP-IPTs, motif 9 and motif 10 could be specifically detected in the Class II CstRNA-IPTs (CsIPT9.1 and CsIPT9.2), whereas no particular motifs could be identified in the Class I CstRNA-IPT CsIPT2 (Figure 3a). The above results revealed that three types of CsIPT protein structures are closely related, and the unique conserved motifs suggest the group-specific functions in the same subgroup.

Wang et al. reported that most *ATP/ADP-IPT* genes contain no or only one intron, whereas Class II tRNA-*IPTs* contain many introns [5]. In this study, 0, 1, or 2 introns were found in tea *Cs*ATP/ADP-*IPT* genes, and 3 *Cs*tRNA-*IPTs* all have many introns. The phylogenetic and motif analyses indicate that *ATP/ADP-IPT* genes are derived from Class II *tRNA-IPTs* [5]. For further study, analysis of the relationship between the gene structure and evolution of *CsIPT* family members can be carried out.

### 2.4. The cis-Acting Elements in the Promoter Regions of CsIPT Genes

The *cis*-regulatory elements act as the binding sites for transcription factor and modulate gene expression by controlling the efficiency of gene promoters, thus identifying the targeted *cis*-elements can be helpful for detailed functional studies [1,8,10]. The results showed that the light-responsive *cis*-elements was prominent in all elements, and they were common to the promoter regions of all *CsIPT* family members. They may control plant endogenous CK level in response to different light conditions. The second are hormone-responsive elements, and the presence of these elements indicates the potential influence of various hormones on the expression of *CsIPT* genes. Abscisic acid (ABA)-responsive elements were present on the promoters of *CsIPT3.1*, *CsIPT3.2*, and *CsIPT7.2*, and they regulate their expression in response to ABA, drought, or salt signals. There were three kinds of abiotic stress-responsive elements that were present on *CsIPT* promoters, including drought, anaerobic induction, and low-temperature (Figure 3b).

### 2.5. Expression Pattern Analysis of CsIPT Family Numbers

The gene expression patterns could provide important information to assess their possible functions. Based on RNA-seq data that were downloaded from the SRA database, and through RNA-seq and qRT-PCR analysis, the expression patterns of the *CsIPT* family numbers in several developmental processes and under abiotic stresses of tea plants were investigated.

#### 2.5.1. Expression Patterns of CsIPT Genes in Different Tissues and to Different Light Treatments

Tissue-specific gene expression regulates the development of the particular organs or tissues [11]. Previous studies suggested that *IPT* genes can be expressed in all of the plant tissues [3,16]. Through analyzing RNA-seq data in the SRA database, the tissue-specific *CsIPT* expression profiles were analyzed to define their precise function. The results showed that *CsIPT6.3* and *CsIPT6.4* had relatively higher transcript levels in the stem than that in the root, whereas their expression was undetected in the leaf, flower, and seed. Except for that of *CsIPT5.2* and *CsIPT7.1* which were expressed in the root and seed, respectively, their expression levels were all zero in the other four tissues. *CsIPT7.2* was all undetected in the five tested tissues. Except for that *CsIPT9.2* expression levels in the flower were zero, its transcription was undetected in the other four tissues. *CsIPT5.1* transcription successively increased in the flower, stem, seed, root, and leaf. *CsIPT3.1* and *CsIPT3.2* showed similar expression patterns, to be specific, their expression levels both successively increased in the stem, leaf, seed, and root. Moreover, *CsIPT3.1* transcription levels in the flower were significantly higher than that in the other four tissues, and *CsIPT3.2* transcription was undetected in the flower. The expression levels of *CsIPT2* and *CsIPT9.1* in the seeds were the highest compared with that in the other four tissues, and their expression levels in the stem and leaf were significantly higher than that in root and flower, respectively. *CsIPT6.1* and *CsIPT6.2* had similar expression patterns. To be specific, they were both not expressed in the stem and leaf, and their expression levels in the root and flower were all markedly higher than that in the seed (Figure 4a).

From the above results, we can see that firstly, compared with others, *CsIPT5.1* and *CsIPT6.1,2* in the root, *CsIPT6.4* in the stem, *CsIPT5.1* in the leaf, and *CsIPT3.1* and *CsIPT6.1,2* in the flower were highly expressed, respectively. Secondly, Cai et al. reported that *IPT3* was predominantly expressed in the phloem of Arabidopisis and *Jatropha*, and *AtIPT6* was abundant in siliques of *Arabidopsis* [9]. Here, the strongest expression of *CsIPT3.1* was in the flower; the strongest expression of *CsIPT3.2* was in the root and seed. The strongest expression of *CsIPT6.1* and *CsIPT6.2* was in the root and flower, and the strongest expression of *CsIPT6.3* and *CsIPT6.4* were in the root and stem of tea plant. The distinct tissue-specific expression patterns of *CsIPT* members indicated their critical roles in the development of different tissues.

Wang et al. reported that tRNA-*IPTs* are constitutively expressed all over the plant [5]. Some studies also reported that tRNA-*IPT*s were highly expressed in all the tissues of maize [1], rice [8], *A. trichopoda*, and *F. vesca* [5]. However, other studies have suggested that tRNA-*IPT*s were expressed ubiquitously, and their stronger expression was in proliferating tissues of *Arabidopsis* and mature leaves of *Jatropha*, respectively [9,10]. Ghosh et al. reported that tRNA-*IPT*s was expressed more in the root, shoot apical meristems, and leaf primordia [8]. Here, *CsIPT9.2* expression was zero or undetected. *CsIPT2* and *CsIPT9.1* were expressed ubiquitously in the five tissues, and their strongest expression was in the seeds. As tRNA-*IPTs* only catalyze cZ biosynthesis [6], thus cZ might be produced mainly in the seed of tea plants.

The responses of *CsIPT* members to different light treatments were studied through analyzing of the RNA-seq data in the SRA database. The results showed that the expression levels of *CsIPT2*, *CsIPT3.1*, *CsIPT5.2*, and *CsIPT9.1,2* were all undetected. Compared with white light, *CsIPT5.1* expression both markedly decreased after exposure to blue and yellow light, and the expression of *CsIPT6.3* and *CsIPT6.4* both markedly decreased after exposure to blue, purple, and yellow light. There were no significant differences about *CsIPT6.1* expression levels among under purple, blue, and white light, and its expression levels under yellow light were significantly lower than that under the three other kinds of light. *CsIPT6.2* expression levels under purple and blue light were both significantly higher than that under white light, however, its expression levels under yellow light were significantly lower than that under the other three kinds of light. *CsIPT3.2* expression levels have no significant differences between that under white and blue light, however, its expression levels under yellow and purple light were both lower than that under white and blue light (Figure 4b). In a word, compared with white light, the three other kinds of light all depressed the expression of *CsIPT6.3* and *CsIPT6.4*, blue and yellow light both markedly depressed *CsIPT5.1* expression, purple and yellow light both markedly depressed *CsIPT3.2* expression, and yellow light markedly depressed the expression of *CsIPT6.1* and *CsIPT6.2*. On the other hand, blue and purple light both significantly induced the transcription of *CsIPT6.2*.

#### 2.5.2. Involvement of CsIPT Genes in Tea Plant Development

##### The Regulatory Roles of CsIPT Genes in Tea Plant Growth and Development

As is known, perennial plants enter dormancy to tolerate low temperatures in the winter, and there are three kinds of dormancy, namely, para-, endo-, and eco-dormancy. Here, the expression profiles of *CsIPT*s in the lateral buds from endo to flush were studied through analyzing RNA-seq data in the SRA database.

The results showed that in the successive stages of endo, eco, and flush, the expression of *CsIPT3.2* and *CsIPT9.2* were all undetected, and the transcriptional levels of *CsIPT5.2* and *CsIPT7.1,2* were all zero. The expression of *CsIPT2*, *CsIPT3.1*, *CsIPT5.1*, *CsIPT6.1,4*, and *CsIPT9.1* all showed a trend of increasing at first and then fell. *CsIPT6.2* expression in endo was undetected, however, its expression levels significantly decreased in the developmental process from eco to flush. *CsIPT6.3* expression in the flush stage was undetected, however, its expression levels were evidently upregulated from endo to eco (Figure 5a). From the above results, it can be seen that *CsIPT2*, *CsIPT3.1*, *CsIPT5.1*, *CsIPT6.1,3*, and *CsIPT 9.1* play important regulatory roles from endo to eco. *CsIPT6.4* play a key role from endo to flush.

With lateral bud development, the transcriptional levels of *CsIPT2*, *CsIPT6.3* and *CsIPT9.1* were all evidently upregulated, however, *CsIPT9.2* expression was undetected. On the contrary, the transcriptional levels of *CsIPT3.1,2*, *CsIPT6.1,2*, and *CsIPT6.4* all significantly decreased. *CsIPT5.1* was not expressed in early lateral buds, however, it was expressed in the late lateral buds. In the early and late lateral bud, the expression levels of *CsIPT7.1,2* and *CsIPT5.2* were all zero (Figure 5b). The above results suggested that the transcription of *CsIPT2*, *CsIPT5.1*, *CsIPT6.3*, and *CsIPT9.1* may promote the development of lateral bud, whereas the transcription of *CsIPT3.1,2* and *CsIPT6.1,2,4* may inhibit its development.

With the leaf development and senescence, *CsIPT5.1* expression first decreased, increased afterwards, and then decreased again. *CsIPT6.4* expression was first decreased and then increased gradually. During the process of leaf development and maturation, the transcription levels of *CsIPT3.1* and *CsIPT3.2* increased, whereas their transcriptional levels both decreased in the subsequent aging process. *CsIPT6.2* expression was only detectable in the second leaf, however, the transcription of *CsIPT6.1 CsIPT7.1*, and *CsIPT7.2* were all undetected in the leaves of all nodes. The expression levels of *CsIPT5.2* and *CsIPT6.3* both decreased during the leaf development and senescence. With leaf development, the expression of *CsIPT2* and *CsIPT9.1* decreased gradually, and there were no significant differences about their expression levels between that in the mature leaf and old leaf. *CsIPT9.2* expression was undetected in the leaf of the first three stages, and its expression was zero in the old leaf (Figure 5c). The above results suggested that the transcription of *CsIPT3.1,2*, *CsIPT5.1*, and *CsIPT6.4* all play important regulatory roles during leaf maturation and senescence.

In the process of alabastrum development into full bloom, *CsIPT5.2* expression increased at first and then decreased; however, *CsIPT2* and *CsIPT3.1,2* were only expressed in the full bloom. In the process of alabastrum development into a half-opened flower, *CsIPT5.1* expression levels increased; the expression levels of *CsIPT7.1* and *CsIPT7.2* were both zero in the flower of three stages; *CsIPT6.4* expression was undetected in the alabastrum, however, its expression was zero in the flower of the two latter stages. The expression levels of *CsIPT6.1*, *CsIPT6.2*, and *CsIPT6.3* were all undetected in the alabastrum, however, their expression levels were all evidently upregulated during the process of half-opened flower development into full bloom. The expression levels of *CsIPT9.1* and *CsIPT9.2* were both undetected in the flower of the three stages (Figure 5d). The above results suggested that *CsIPT5.1,2* and *CsIPT6.1,2,3* may all play key regulatory roles during alabastrum development into full bloom, whereas *CsIPT2* and *CsIPT3.1,2* may play regulatory roles at the stage of full bloom.

##### CsIPT Numbers Synergistically Regulated the Growth of Leaf-Bud and Flower-Bud Induced by Pruning in Tea Plant

CKs are primarily synthesized in the roots and transported acropetally, and locally synthesized CKs also play key roles in plant development, such as promoting lateral bud outgrowth [4,17]. In the late July, the lateral bud in the leaf axil of tea plant have differentiated into a leaf-bud in the middle and two flower-buds on both sides. Here, the growth dynamic of the flower-bud and leaf-bud on the first node below the cuttings was studied within 14 days after pruning. From the dynamic changes of the relative size and organizational structure of the leaf-bud and flower-bud, it can be seen that compared with control, pruning promoted the rapid growth of the leaf-bud, whereas depressing the growth rate of the flower-bud and did not cause its abnormality. Over time, after 6 months of pruning, most of the original flower-buds below the cuttings dried, rot, or fell off, and a few remaining flower-buds developed slowly or stopped development, whereas the leaf-buds have developed into the new lateral branches (Figure 6a,b). Thus, the above results showed that pruning depressed flower-bud growth and promoted leaf-bud growth simultaneously.

The qRT-PCR results showed that, in the leaf-bud after pruning, the expression levels of *CsIPT3.1,2* and *CsIPT5.1* all significantly decreased at 3d and 7d. The expression of *CsIPT5.2* and *CsIPT7.1* both significantly decreased at 3d and increased at 7d. The expression of *CsIPT6.1* and *CsIPT6.2* both significantly decreased at 3d. At 3d and 7d, pruning induced the transcription of *CsIPT6.3* and *CsIPT6.4*, respectively. Compared with the control, *CsIPT7.2* expression showed a trend of decrease, increase, and decrease at 3d, 7d, and 14d, respectively (Figure 6c–o). In a word, the above results showed that from 0 to 14d after pruning, pruning may promote the growth of the leaf-bud through inducing the transcription of *CsIPT5.2*, *CsIPT6.3,4*, and *CsIPT7.1,2* in the leaf-bud at different time points.

On the other hand, the qRT-PCR results showed that in the flower-bud, pruning significantly induced the expression of *CsIPT3.1* and *CsIPT5.1,2*, and significantly inhibited the expression of *CsIPT6.1,4* simultaneously. *CsIPT3.2* expression significantly decreased at 3d, however it significantly increased at 7d. At 3d, 7d, and 14d after pruning, *CsIPT6.2* expression decreased firstly, then increased, and decreased finally. *CsIPT7.1* expression levels showed a trend of decreasing at 3d and then increasing and 7d after pruning. *CsIPT7.2* expression was downregulated at 3d, however it was evidently upregulated at 7d and 14d subsequently (Figure 7a–m). In a word, the qRT-PCR results showed that pruning may depress the growth of flower-bud through inhibiting the transcription of *CsIPT3.2*, *CsIPT6.1,2,4*, and *CsIPT7.1,2* in the flower-bud at different time points.

The RNA-seq results showed that after 3d of pruning, compared with the control, the expression levels of *CsIPT3.2* in the flower-bud significantly decreased, and there were no significant differences about its expression between the control and pruning in the leaf-bud. *CsIPT6.1* expression in the leaf-bud significantly increased and that in the flower-bud significantly decreased. Compared with the control, the expression of *CsIPT3.1* and *CsIPT5.1* in the flower-bud and leaf-bud both significantly decreased, and the decreased degree of their expression in the flower-bud was significantly higher than that in the leaf-bud. *CsIPT6.2* expression levels in the flower-bud significantly decreased, however, its transcription in the leaf-bud was undetected; *CsIPT6.4* expression in flower-bud and leaf-bud were both undetected. The expression levels of *CsIPT5.2*, *CsIPT6.3*, and *CsIPT7.1,2* in flower-bud and leaf-bud were all zero (Figure 7n).

Here, *CsIPTs*’ expression levels in the leaf-bud and flower-bud at 3d after pruning were detected by qRT-PCR and RNA-seq simultaneously, and the similarities and differences between the two results were compared. Firstly, the results of qRT-PCR and RNA-seq suggested that pruning induced the transcription of *CsIPT6.1* and *CsIPT6.3* in the leaf-bud, respectively. Although the results were inconsistent, *CsIPT6.1* and *CsIPT6.3* are homologous genes and they may have similar functions. On the contrary, the two results both suggested that pruning depressed the transcription of *CsIPT3.1* and *CsIPT5.1* in the leaf-bud. The results of qRT-PCR also suggested that pruning depressed the transcription of *CsIPT3.2*, *CsIPT5.2*, *CsIPT6.1,2*, and *CsIPT7.1,2* in the leaf-bud. Secondly, the two results both suggested that pruning depressed the transcription of *CsIPT3.2* and *CsIPT6.1,2* in the flower-bud. The qRT-PCR results suggested that pruning depressed the transcription of *CsIPT6.4* and *CsIPT7.1,2*, whereas the RNA-seq results showed that pruning depressed the transcription of *CsIPT3.1* and *CsIPT5.1* in the flower-bud.

In summary, firstly, the qRT-PCR results showed that *CsIPT6.1* transcription may depress flower-bud growth. The transcription of *CsIPT5.2* and *CsIPT6.3* may promote leaf-bud growth. The transcription of *CsIPT6.4* and *CsIPT7.1,2* may depress flower-bud growth and promote leaf-bud growth simultaneously. Secondly, the RNA-seq results showed that *CsIPT6.1* transcription may promote leaf-bud growth and depress flower-bud growth simultaneously. The transcription of *CsIPT3.1* and *CsIPT5.1* both depressed the growth of the leaf-bud and flower-bud, and the depressed degree of the flower-bud growth may be both higher than that of the leaf-bud. Thirdly, the two results both showed that the transcription of *CsIPT3.2* and *CsIPT6.2* both depressed flower-bud growth. These results warrant further investigation in order to understand the roles of *CsIPTs* in synergistically regulating the growth of leaf-buds and flower-buds.

#### 2.5.3. The Expression Patterns of CsIPT Genes in Response to Abiotic Stresses

Abiotic stresses can cause irreversible injury to the growth and development of tea plants [11]. The gene expression patterns in response to abiotic stresses suggested their multiple potential functions. Previous studies showed that ATP/ADP-*IPT* genes are involved in plant responses to abiotic stresses, whereas the transcription of tRNA-*IPT* genes have relatively little variation when plants are subjected to abiotic stresses [5]. For example, the expression levels of three ATP/ADP-*IPT* genes in angiosperms are substantially reduced when plants are subject to heat-, cold-, drought-, or salt-stress [5]. However, other studies also reported that tRNA-*IPT* genes also have important functions in the adaptation to abiotic stress [10]. Here, the expression patterns analysis of *CsIPT* genes in response to several abiotic stresses was performed based on publicly available transcriptome data.

##### The Involvement of CsIPT Family Members in Response to Temperature Stresses

In the control and cold-stress treatments, the expression levels of six *CsIPT* genes (*CsIPT6.1,2,3,4* and *CsIPT7.1,2*) all were zero. Compared with the control, *CsIPT5.2* expression levels under 4 °C for 8 h significantly increased, whereas its expression did not change under the other three cold-stress conditions. Compared with the control, the expression levels of *CsIPT2*, *CsIPT3.1*, *CsIPT5.1*, and *CsIPT9.1* all significantly decreased after cold-stress treatments. The expression of *CsIPT3.2* was undetectable after cold-stress for 4 h, whereas its expression levels significantly decreased after cold-stress for 8 h compared with the control. *CsIPT9.2* expression was undetected in all the treatments (Figure 8a). In a word, under 4 °C for 8 h, the transcription of *CsIPT5.2* may play a key role in the response to cold-stress of tea plants. On the other hand, compared with the control, the expression of *CsIPT2*, *CsIPT3.1,2*, *CsIPT5.1*, and *CsIPT9.1* all significantly decreased after cold-stress treatments.

Compared with the control, 4 °C and 0 °C both decreased the expression of *CsIPT2*, *CsIPT3.1*, *CsIPT5.1*, and *CsIPT9.1* in mature leaves and young leaves, and decreased the expression of *CsIPT3.2* and *CsIPT6.1* in young leaves. The expression of *CsIPT3.2* and *CsIPT6.1* in mature leaves were both undetected. The expression of *CsIPT6.4* and *CsIPT9.2* in mature leaves and young leaves were all undetected. For the cold treatments and control, the expression levels of *CsIPT7.1,2* in the young leaves were all 0, and their expression in the mature leaves were undetected. For the control and cold treatments, *CsIPT5.2* expression levels in the young leaves were all 0, whereas cold treatments decreased *CsIPT5.2* expression levels in the mature leaves compared with the control. *CsIPT6.3* expression was undetected in mature leaves under 0 °C, whereas 4 °C and 0 °C both decreased its expression in young leaves and 4 °C decreased its expression in mature leaves. 4 °C and 0 °C both decreased the *CsIPT6.2* expression levels in young leaves, whereas 4 °C enhanced its expression and 0 °C significantly decreased its expression in mature leaves compared with the control (Figure 8b). In a word, under 4 °C, *CsIPT6.2* may play a key regulatory role for increased resistance to cold-stress in mature leaves. 4 °C and 0 °C both decreased the expression of *CsIPT2*, *CsIPT3.1*, *CsIPT5.1*, and *CsIPT9.1* in mature leaves and young leaves and decreased the expression of *CsIPT3.2* and *CsIPT6.1,2,3* in young leaves.

In all the controls and treatments, the expression of *CsIPT6.2*, *CsIPT7.1,2*, and *CsIPT9.1* were all undetected. Firstly, under ML, the expression of *CsIPT2*, *CsIPT3.2*, *CsIPT5.1*, *CsIPT6.3*, and *CsIPT9.2* decreased, the expression levels of *CsIPT3.1*, *CsIPT5.2*, and *CsIPT6.1* were unchanged, and *CsIPT6.4* expression levels were undetected. Secondly, under SL, *CsIPT6.3* expression levels were undetected, the expression of *CsIPT2*, *CsIPT3.2*, *CsIPT5.1*, *CsIPT6.1*, and *CsIPT9.2* decreased, whereas the expression of *CsIPT3.1*, *CsIPT5.2*, and *CsIPT6.4* unchanged. Thirdly, under MH, the expression of *CsIPT3.1* and *CsIPT6.3.4* were undetected, the expression of *CsIPT2*, *CsIPT3.2*, *CsIPT5.1*, *CsIPT6.1*, and *CsIPT9.2* all significantly decreased, and *CsIPT5.2* expression unchanged. Fourthly, under SH, the expression of *CsIPT6.3* and *CsIPT6.4* were undetected, the expression of *CsIPT3.1*, *CsIPT5.2*, and *CsIPT6.1* did not change, the expression of *CsIPT2*, *CsIPT5.1*, and *CsIPT9.2* significantly decreased, however, *CsIPT3.2* expression significantly increased (Figure 8c). Therefore, the enhanced transcriptional levels of *CsIPT3.2* may play a key regulatory role in improving the resistance to SH of tea plants. On the other hand, ML significantly repressed the expression of *CsIPT2*, *CsIPT3.2*, *CsIPT5.1*, *CsIPT6.3*, and *CsIPT9.2*; SL significantly depressed the expression of *CsIPT2*, *CsIPT3.2*, *CsIPT5.1*, *CsIPT6.1*, and *CsIPT9.2*; MH significantly repressed the expression of *CsIPT2*, *CsIPT3.2*, *CsIPT5.1*, *CsIPT6.1*, and *CsIPT9.2*; and SH significantly repressed the expression of *CsIPT2*, *CsIPT5.1*, and *CsIPT9.2*.

The age of tea trees that were used in the three transcriptomes is different. The cultivar that was used in Figure 8a,b is LJ43 and in Figure 8c is SCZ, which has stronger tolerance under cold stress. The whole tea plants were exposed to temperature stress in Figure 8a, whereas the branches that were cultured in vitro were exposed to temperature stress in Figure 8b. In Figure 8a, the mature leaves were collected at 4 h and 8 h. In Figure 8b, the mature leaves and young leaves were collected at 4 h simultaneously. In Figure 8c, the mature leaf was collected at 4d and 8d for ML and SL, respectively.

In summary, the transcription of *CsIPT6.2* and *CsIPT5.2* in the mature leaves may enhance the tolerance of tea plants to cold stress under 4 °C for 4 h and 8 h, respectively, and *CsIPT3.2* transcription in the mature leaves may enhance its tolerance under SH. On the contrary, −5 °C and 4 °C for 8 h as well as ML, SL, and MH all markedly decreased *CsIPT3.2* expression in the mature leaves. From Figure 8a,b, we can see that compared with control, the expression levels of *CsIPT2*, *CsIPT3.1*, *CsIPT5.1*, and *CsIPT9.1* in the mature leaves all decreased under 5 °C, 0 °C, and 4 °C for 4 h as well as under −5 °C and 4 °C for 8 h, and their expression levels in young leaves both decreased under 4 °C and 0 °C for 4 h. In addition, 4 °C and 0 °C for 4 h also both decreased the expression of *CsIPT3.2* and *CsIPT6.1,2,3* in young leaves. On the other hand, ML, SL, MH, and SH all significantly depressed the expression of *CsIPT2*, *CsIPT5.1*, and *CsIPT9.2* in mature leaves; SL and MH both significantly repressed *CsIPT6.1* expression in mature leaves; ML significantly repressed *CsIPT6.3* expression in mature leaves.

##### The Involvement of CsIPT Family Members in Response to Drought and Salt Stresses

The expression of *CsIPT3.1* and *CsIPT6.4* was undetected in the control, however, their expression levels were zero in the drought and rewatering treatments. In the control, drought, and rewatering treatments, the expression levels of *CsIPT5.2*, *CsIPT6.1,2*, and *CsIPT7.1,2* were all zero. Compared with the control, *CsIPT3.2* transcriptional levels in drought and rewatering both significantly increased, and its transcriptional levels were higher after rewatering. Compared with the control, *CsIPT5.1* expression significantly decreased after drought treatment, however, its expression was undetectable after rewatering. Compared with the control, *CsIPT6.3* expression significantly decreased after drought treatment; moreover, its expression continued to decrease after rewatering. The expression levels of *CsIPT2* and *CsIPT9.2* were undetected in all the treatments. *CsIPT9.1* expression was zero under drought-stress, whereas its expression levels were both undetected in the other two treatments (Figure 9a).

*CsIPT5.1* expression was undetected in the controls, however, its expression levels were zero under salt-stress. In control and salt-stress, the expression levels of *CsIPT5.2*, *CsIPT6.1,2*, and *CsIPT7.1,2* were all zero. Compared with the control, the expression of *CsIPT3.1,2* and *CsIPT6.3,4* significantly decreased after salt treatment. The expression levels of *CsIPT9.1* and *CsIPT9.2* were both undetected in the control, while their expression was both zero after salt treatment. *CsIPT2* expression was undetected in both the control and salt treatment (Figure 9b).

The structural divergence of gene sequences plays vital roles in the adaptation of plant to abiotic stresses [11]. Ghosh et al. reported that the structure and function of *IPT3/5/7* genes are more closely related to each other than other ADP/ATP-*IPT* genes, and cold-stress upregulates *IPT3/5/7* transcription whereas down-regulates the rest [8]. In *B. rape*, three tRNA-*IPT* genes have unique gene structures and expression patterns [10]. In this study, the homologous genes with similar gene sequences sometimes have similar expression patterns. For example, *CsIPT3.1* and *CsIPT3.2* had similar expression patterns in different tissues, as well as during the processes of leaf maturation and aging, lateral bud development, and flower development. Likewise, *CsIPT6.1* and *CsIPT6.2* had similar expression patterns in the five tissues, and *CsIPT6.1*, *CsIPT6.2*, and *CsIPT6.4* all markedly decreased in the process of lateral bud development. *CsIPT6.1*, *CsIPT6.2*, and *CsIPT6.3* had similar expression profiles during the process of flower development. During the process of dormancy and flush of lateral bud, *CsIPT7.1* and *CsIPT7.2*, *CsIPT6.1* and *CsIPT6.4* had similar expression profiles, respectively.

After pruning, the expression patterns of *CsIPT3.1* and *CsIPT3.2*, *CsIPT6.1* and *CsIPT6.2* in the leaf-bud were similar, respectively, and the expression patterns of *CsIPT5.1* and *CsIPT5.2* in the flower-bud were similar. After pruning, *CsIPT7.1* and *CsIPT7.2* had similar expression patterns both in the leaf-bud and flower-bud. Under cold-stress (4 °C and −5 °C), *CsIPT6.1*, *CsIPT6.2*, *CsIPT6.3*, and *CsIPT6.4* have similar expression patterns. After cold, low-temperature, and high-temperature treatments, *CsIPT7.1* and *CsIPT7.2* in young leaves and mature leaves all have similar expression patterns. Furthermore, they have similar expression patterns under cold-stress (4 °C and −5 °C) as well as during the process of flower development. Under different light treatments, MH and SH, *CsIPT6.3* and *CsIPT6.4* all had similar expression profiles. *CsIPT9.1* and *CsIPT9.2* had similar expression profiles under different light treatments as well as during the process of flower development.

In summary, the above results suggested that *CsIPT3.2* played key regulatory roles in enhanced resistance to drought-stress and rewatering of tea plants. Drought treatment depressed *CsIPT5.1* transcription, whereas drought and rewatering both depressed *CsIPT6.3* transcription. On the other hand, salt-stress significantly depressed the expression of *CsIPT3.1,2* and *CsIPT6.3,4*. We can see that the expression patterns of tea plant *CsIPT* numbers depend on the type and degree of the abiotic stress treatment, which is consistent with previous reports. Ghosh et al. reported that different members of the *AtIPT* family responded differently depending on the type of stress exposure [8]. The expression of *OsIPT5* markedly increased in response to salt, dehydration and oxidative stresses [8]. Many types of transgenic plants with pSARK::IPT construct exhibited enhanced resistance to drought-stress, whereas the resistance of cotton plants did not change under salt-stress [18]. Salt-stress suppressed, whereas cold- or drought-stress enhanced the expression of most *ATP/ADP-IPT* genes in rice [8]. *Arabidopsis* quadruple ADP/ATP-*IPT* mutant *ipt1;3;5;7* display increased tolerance under drought and salt conditions [19].

Le et al. reported that *GmIPT9* and *GmIPT13* in soybean were significantly induced by drought and the induced degree was higher in the younger trifoliate leaves than in the V6-stage leaves [20]. Here, the results showed that under 4 °C and 0 °C, there are more *CsIPT* genes in young leaves whose expression levels were depressed compared with that in mature leaves, and this may be one of the most important reasons why young leaves have relative resistance compared with mature leaves. On the other hand, Wang et al. reported that *ATP/ADP-IPTs* regulates organ development and stress responses, while tRNA-*IPTs* mainly plays housekeeping roles in angiosperms [5]. Here, the results showed that *ATP/ADP-CsIPTs* and tRNA-*CsIPTs* were both involved in the processes of growth, development, and abiotic stress response, which is consistent with other previous studies that tRNA-*IPTs* are also involved in stress response. For example, Ghosh et al. reported that transcript abundance of *OsIPT2* and *OsIPT9* fluctuated after stress treatment [8], and Liu et al. reported that the tRNA-*IPT* genes notably responded to adverse abiotic stresses in *B. rape* [10].

Although the stress response *cis*-elements are present in the promoter sequences of *CsIPT* genes, it is difficult to make direct connections between these elements and the expression patterns in response to a specific abiotic stress. Therefore, further study to confirm the expression patterns of *CsIPT* genes in stress factor-treated tea plants is needed.

In summary, the results here showed that firstly, *CsIPT5.1* and *CsIPT6.1,2* in root, *CsIPT6.4* in stem, *CsIPT5.1* in leaf, *CsIPT3.1* and *CsIPT6.1,2* in flower, and *CsIPT2* and *CsIPT9.1* in seed were highly expressed, respectively. Secondly, compared with white light, blue and purple light both significantly induced *CsIPT6.2* expression. *CsIPT6.4* may play a key role during the dormancy and flush of lateral bud. *CsIPT2*, *CsIPT5.1*, *CsIPT6.3*, and *CsIPT9.1* may promote lateral bud development. *CsIPT3.1,2*, *CsIPT5.1*, and *CsIPT6.4* may play key roles during leaf development. *CsIPT5.1,2* and *CsIPT6.1,2,3* may play key roles during flower development. *CsIPT3.2*, *CsIPT5.2*, *CsIPT6.1,2,3,4*, and *CsIPT7.1,2* may play key regulatory roles in the synergistic growth of leaf-bud and flower-bud that are induced by pruning. Thirdly, *CsIPT5.2* may play a key role in response to 4 °C for 8 h. *CsIPT6.2* may play a key regulatory role for increased resistance to 4 °C cold-stress in mature leaves. *CsIPT3.2* may play a key role in improving the resistance to severely high-temperature stress. *CsIPT3.2* may also play key roles in enhanced resistance to drought-stress and rewatering.

## 3. Materials and Methods

### 3.1. Identification of CsIPT Genes in Tea Plants

Using the amino acid sequences of *Arabidopisis*, AtIPT as queries (the detailed information is shown in Appendix A), the local BLASTP was executed to search the IPT proteins in two tea plant genome databases [5,21]. A total of nine and seven *CsIPT* genes were retrieved from the genome database of cv. SCZ and cv. LJ43, respectively. Next, CsIPT sequences were submitted to the NCBI CD-search program (https://www.ncbi.nlm.nih.gov/Structure/cdd/wrpsb.cgi accessed on 16 October 2020) to obtain the conserved domains in the candidate proteins, and TBtools software was used for visualization [22]. All of the obtained putative CsIPT protein sequences were submitted to the Pfam database (http://pfam.janelia.org/ accessed on 16 October 2020), the Conserved domain database (http://www.ncbi.nlm.nih.gov/Structure/cdd/wrpsb.cgi accessed on 16 October 2020), and SMART (http://smart.embl-heidelberg.de/ accessed on 16 October 2020) to confirm the presence of domain signatures.

### 3.2. Multiple Sequence Alignment and Phylogenetic Analysis

The full-length amino acid sequences of 13 *C. sinensis* CsIPT and 9 *A. thaliana* AtIPT proteins were used to explore the corresponding phylogenetic relationships. Multiple sequence alignments of the identified CsIPTs with their *A. thaliana* homologues were performed using MUSCLE in MEGA 7.0. The alignment was conducted using 22 IPT proteins. Manual adjustments were carried out when needed. A phylogenetic tree was constructed with the 1000 bootstrapped maximum likelihood (ML) method. The best-fit model for protein evolution was selected. The subfamilies are divided based on cluster analysis [1,10].

### 3.3. Analysis of the Gene Structure and Conserved Motifs of CsIPT Genes

To explore the gene structural compositions and the conservation of the *CsIPT* genes, the exon-intron structures were identified via uploading the GFF3 file to the TBtools program. The conserved motifs in the *CsIPT* genes were screened and identified using the MEME online tool (http://meme-suite.org/tools/meme accessed on 16 October 2020), with a maximum number of 10 motifs, and then graphically displayed by the TBtools software. The TBtools program was used to combine the results of conserved motifs or conserved domains with a phylogenetic tree into one map, respectively [22].

### 3.4. Physiological and Biochemical Property Analysis of CsIPT Proteins

The theoretical isoelectric points (pIs), molecular weights (MWs), and grand average of hydropathy (GRAVY) of CsIPT proteins were predicted using the ProtParam tool (http://web.expasy.org/protparam/ accessed on 16 October 2020). The glycosylation sites were predicted using NetNGly (http://www.cbs.dtu.dk/services/NetNGlyc/ accessed on 16 October 2020). Subcellular localization and the number of transmembrane domains were predicted with WOLF PSORT (https://wolfpsort.hgc.jp/ accessed on 16 October 2020) and TMHMM Server (version 2.0) (http://www.cbs.dtu.dle/services/TMHMM/ accessed on 16 October 2020), respectively. The CsIPT sequences were analyzed with ExPASyProtParam (http://www.expasy.org/tools/protparam.html accessed on 16 October 2020) to obtain the number of amino acids and instability index. The Signal peptides of CsIPT proteins were analyzed using the SignalP 4.1 Server (http://www.cbs.dtu.dk/services/SignalP/ accessed on 16 October 2020).

### 3.5. Promoter Cis-Acting Regulatory Elements Analysis

The upstream genomic sequences within 2000 bp of the start codon of all the *CsIPT* genes were extracted from the genome. The *cis*-acting elements in the putative promoter regions were identified using PlantCARE database (bioinformatics.psb.ugent.be/webtools/plantcare), and the identified *cis*-acting elements were classified by their different functions and visualized using TBtools software [22].

### 3.6. Publicly Available Transcriptome Data Analysis

To explore the expression profiles of *CsIPT* genes, the raw RNA-seq data were downloaded from the NCBI Short-Read Archive (SRA) database (https://blast.ncbi.nlm.nih.gov/Blast.cgi accessed on 16 October 2020).

#### 3.6.1. Analysis of *CsIPT* Expression Patterns in SRA Database during Tea Plant Development

A total of four microarray datasets were downloaded for studying their expression profiles in different tissues under different light treatments, and during tea plant development. (1) transcriptional levels of 4-year-old tea cv. LJ43 in different tissues, during lateral bud development, and leaf development and senescence. The lateral buds, first leaves, second leaves, and stems were collected on April 6; mature leaves, old leaves, and roots were collected on June 27; and the flowers and seeds were collected on November 17 (accession number SRP034436) [23]; (2) transcriptional levels of ten-year-old tea cv. ZC108 that were exposed to different light treatments. Tea bushes were covered with blue film, purple film, yellow film, and colorless transparent film (control) before the tea plants sprouted in the spring, respectively. One leaf and one bud were harvested when the first leaf was fully unfolding (SRR4255671 to SRR4255674) [24]; (3) the transcriptional levels of 10-year-old cv. LJ43 during dormancy and bud flush (flush) of lateral bud. The lateral buds at the stage of endodormancy (endo), ecodormancy (eco), and flush were sampled at 1 December 2013, 14 February 2014, and 14 March 2014, respectively (SRR5040773 to SRR5040784) [25]; (4) the transcriptional levels of cv. Ruixue during flower development. The flowers at three opening stages were harvested on the same day (SRR5487527 to SRR5487532) [26].

#### 3.6.2. Analysis of *CsIPT* Gene Expression in the SRA Database during Abiotic Stress

There were three microarray datasets of temperature stresses in the tea plant that were used to analyze the expression of *CsIPT* genes. (1) The transcriptional levels of two-year-old cv. LJ43 under chilling and freezing stresses. The third leaf was harvested (SRP051838) [27]; (2) transcriptional levels of 16-year-old cv. LJ43 under cold stresses. The branches that were cultured in vitro were treated at 20 °C (control), 4 °C, and 0 °C for 4 h. The mature leaves (the third and/or fourth leaves below the apical bud) and young leaves (the two leaves and a bud) were collected separately (SRP116862) [16]; (3) transcriptional levels of 1-year-old cv. SCZ under high- and low-temperature stresses. The tea plants were moved into a growth chamber with a 16 h light (25 °C)/8 h dark (20 °C) photoperiod with 3600 Lx photos m^−2^ s^−2^ light intensity and 75% humidity, for 2 weeks before treatments. For the low-temperature treatments, the tea plants were exposed to 18/15 °C for 2d, 10/8 °C for 2d, 4/2 °C for 2d, and 2/0 °C for 2d. The third leaf at the fourth day (moderately low-temperature stress, ML) and eighth day (severely low-temperature stress, SL) was collected. For high-temperature stresses, the tea plants were exposed to 28/26 °C for 2d, 30/27 °C for 2d, 36/30 °C for 2d, and 39/32 °C for 2d. The third leaf at the fourth day (moderately high-temperature stress, MH) and eighth day (severely high-temperature stress, SH) was collected, respectively. The third leaf that was grown under normal conditions (25/20 °C) at the fourth day (control M, CKM) and eighth day (control S, CKS) was collected, respectively (accession number SRP116862) [28].

There were two microarray datasets that were used to analyze *CsIPT* expression under drought and salt stresses. (1) The transcriptional levels of 10-year-old cv. Ningzhou2 under drought-stress and recovery. The tea plants were subjected to severe drought stress and then allowed to recover after rewatering in field conditions, and the ‘two and a bud’ samples (one young shoot with two leaves and a bud) were collected (PRJNA297732) [16]; (2) the transcriptional levels of one-year-old cv. Pingyangtezao under salt-stress. The cutting seedlings that were incubated in vitro were treated with 250 mM NaCl for 4 h, and the first and second leaves were collected (SRP107589) [29].

The quick run trimmomatic plugin of TBtools software was used to filter low-quality transcriptome raw sequences. These downloaded RNA-seq data were remapped back to the reference genome of LJ43 and SCZ, respectively, when needed. The transcripts per kilobase million (TPM) values were standardized by the kallisto super GUI wrapper plugin of the TBtools software, and the expression levels of the *CsIPT* genes were plotted on a log2 scale. Heat maps were established based on the TPM value of each *CsIPT* gene using the TBtools software [22].

### 3.7. RNA-seq and qRT-PCR Detection

For the pruning experiment, the adult tea plants of cv. ZC108 which were heavily pruned in late April were used. The experiment was conducted in the tea garden of Shengzhou experiment base, Tea Research Institute, Chinese Academy of Agricultural Sciences, located in Shaoxing, Zhejiang Province, China. The summer pruning was conducted at a height that was 20 cm higher than the last cut surface on 27 July 2021. The control tea bushes were not pruned. At 3d, 7d, and 14d after pruning, the leaf-buds and flower-buds of the first node below the cuttings were sampled, frozen into liquid N_2_ immediately and stored at −80 °C. The leaf-buds and flower-buds were collected from tea bushes of 2 m long as one biological replicate, and four biological replicates were sampled and measured for each treatment.

Total RNA was extracted using the previously described CTAB method for Illumina RNA-seq and qRT-PCR testing. The extracted RNA was treated with DNase I (Invitrogen, CA, USA). The integrity, concentration, and purity of all the RNA samples were analyzed and monitored. cDNA library construction and sequencing were performed by Novogene Co., Ltd. (Beijing, China) [24]. The reverse transcription and qRT-PCR assay were performed according to our previous study [14].

### 3.8. The Preparation and Observation of Paraffin Sections

The axillary buds were cut out and immersed into FAA fixation fluid immediately. Aspirating twice for 15 min each time, and fixation at room temperature for more than 24 h. Subsequently, they were subjected to gradient dehydration through alcohol and tert-butanol. An equal volume of melted paraffin was added into the sample/tert-butanol, and then the sample was placed at 60–65 °C for more than 24 h. The sample with melted pure paraffin was poured into the carton. When the wax solidified slightly, it was placed into normal temperature water immediately. The wax block was fixed and cut into slices, heating the sections at 45 °C overnight. Dewaxing and staining was undertaken with TBO. The sheet was sealed and it was photographed it with microscope.

### 3.9. Statistical Analysis

The data were analyzed using Statistica (SAS Institute, Inc., Cary, NC, USA, http://www.statsoft.com accessed on 16 October 2020). In each figure, differences of gene expression between the control and pruning treatment were determined by Student’s *t*-tests.

## 4. Conclusions

Based on highly conserved gene structures and motifs, 13 *CsIPT* genes were identified from the tea plant genome of two cultivars, and they were phylogenetically divided into four subfamilies. Moreover, 10 *CsIPT* members belonged to plant ADP/ATP-*IPT* genes, and 3 *CsIPTs* were tRNA-*IPT* genes. All of the CsIPTs showed conserved motif structures among the members of every type, and there is a conserved putative ATP/GTP-binding site (P-loop motif) in all the CsIPT sequences.

Through analyzing the gene expression patterns, *CsIPT* genes which play key regulatory roles in the different tissues of tea plants were screened, respectively. Furthermore, it can be seen that *CsIPT6.2* may be involved in the response to different light treatments. *CsIPT6.4* may play a key role during the dormancy and flush of lateral buds. *CsIPT5.1* may all play key regulatory roles in the developmental processes of the lateral bud, leaf, and flower. *CsIPT5.2* and *CsIPT6.2* may both play key regulatory roles for increased resistance to 4 °C cold-stress, whereas *CsIPT3.2* may all play key roles in improving resistance to severely high-temperature, drought-stress and rewatering.

This study could provide important information for further functional studies and molecular breeding of tea plants. For further study, such as using expression of *CsIPT* promoter-reporter fusion genes and in situ hybridization technique to confirm their expression patterns in tea plants is needed.

## Figures and Tables

**Figure 1 plants-11-02243-f001:**
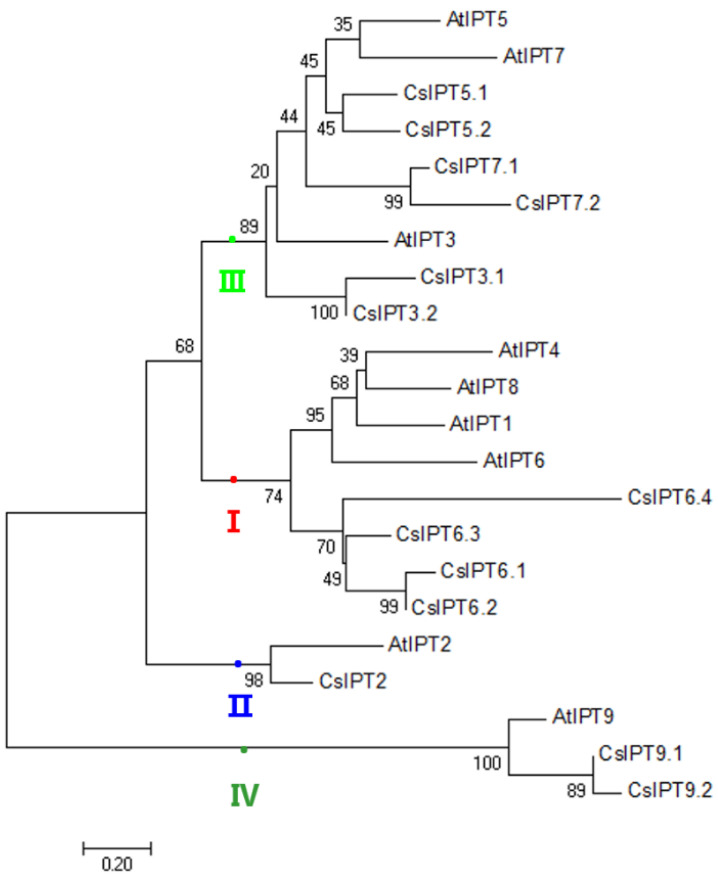
Phylogenetic relationships of *CsIPTs*, together with their *Arabidopsis* counterparts, respectively. The trees were constructed using the NJ method by the MEGA (version 7.0) program. *CsIPTs* were divided into four groups indicated by differently colored dots. The roman figures I, II, III, and IV respresent each subgroup that *CsIPTs* and *AtIPTs* were clustered, respectively.

**Figure 2 plants-11-02243-f002:**
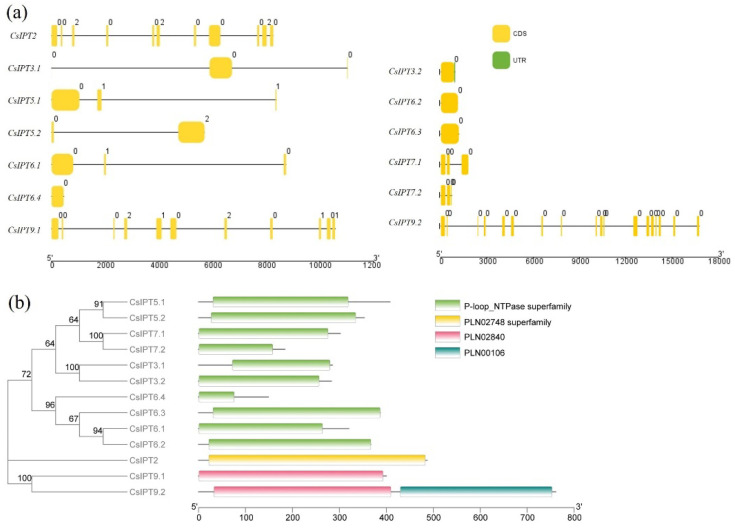
Gene structure and conserved domain analysis of CsIPTs. The exons and introns were represented by brown boxes and black lines (**a**). Conserved domain analysis of CsIPTs at the evolutionary level. Phylogenetic tree and conserved domains were combined into one map using the TBtools program. The relative positions of each domain within each protein were shown in color (**b**).

**Figure 3 plants-11-02243-f003:**
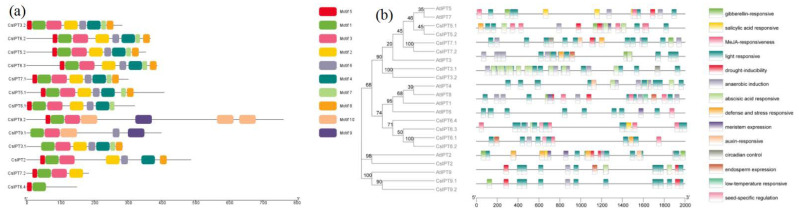
Conserved motifs and *cis*-element analysis of *CsIPTs*. Conserved motifs analysis of *CsIPTs* (**a**). Plant CARE was used to identify the putative *cis*-acting element distribution in 2000 bp promoter sequences of 13 *CsIPT* genes. The phylogenetic tree and the *cis*-element were combined into one map using the TBtools program (**b**).

**Figure 4 plants-11-02243-f004:**
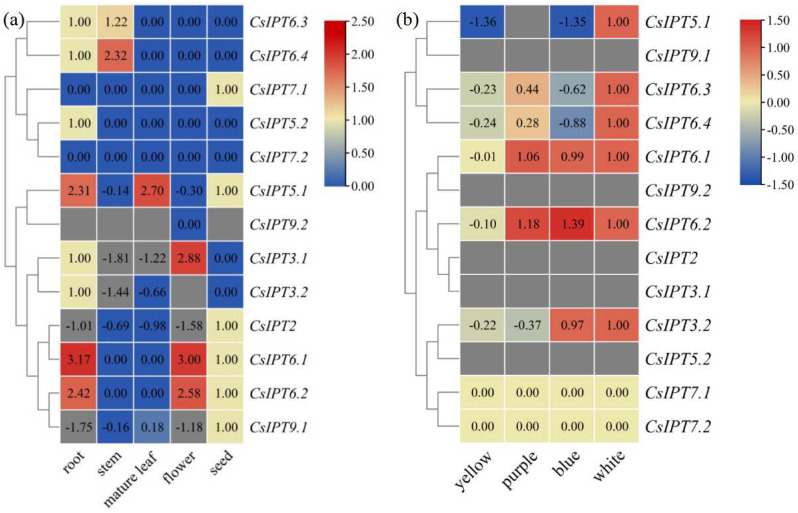
Expression profiles of *CsIPT* genes in different tissues (**a**) and to different light treatments (**b**) in tea plants.

**Figure 5 plants-11-02243-f005:**
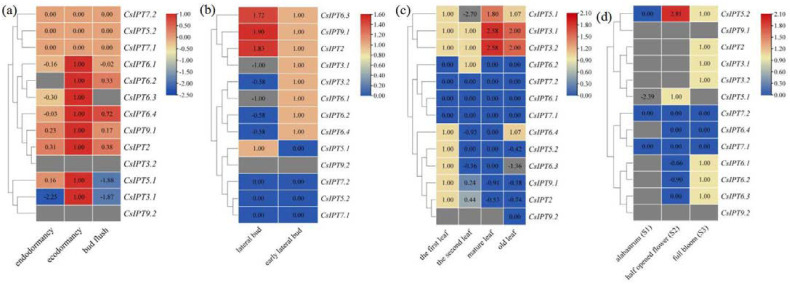
Expression profiles of *CsIPT* genes in different development stages of tea plants. (**a**) The expression profiles of *CsIPTs* during the stages from endodormancy to bud flush of lateral buds. (**b**) The expression profiles of *CsIPTs* during the process of lateral bud development. (**c**) The expression profiles of *CsIPTs* during the process of leaf development and senescence. (**d**) The expression profiles of *CsIPTs* during the process of flower development.

**Figure 6 plants-11-02243-f006:**
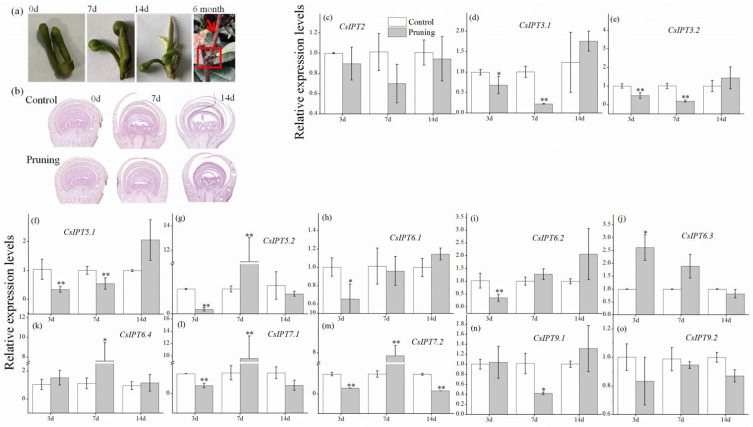
The photographs and paraffin sections of the lateral bud during the process of lateral bud development that was induced by pruning (**a**,**b**). Expression profiles of *CsIPT* genes in the leaf-bud that were induced by pruning (**c**–**o**). In each figure, asterisks show marked difference in the expression levels of each *CsIPT* gene between control and pruning (* *p* < 0.1; ** *p* < 0.01; Student’s *t*-test).

**Figure 7 plants-11-02243-f007:**
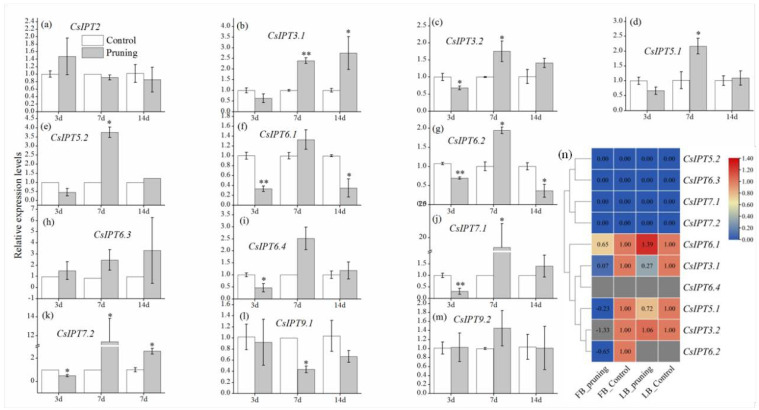
Expression profiles of *CsIPT* genes in the flower-bud and leaf-bud that were induced by pruning. The expression profiles of *CsIPTs* in the flower-bud that were detected by qRT-PCR at different time points after pruning (**a**–**m**). The expression profiles of *CsIPTs* in the flower-bud and leaf-bud that were detected by RNA-seq at different time points after pruning (**n**). In each figure, asterisks show marked difference in the expression levels of *CsIPT* gene between control and pruning (* *p* < 0.1; ** *p* < 0.01; Student’s *t*-test).

**Figure 8 plants-11-02243-f008:**
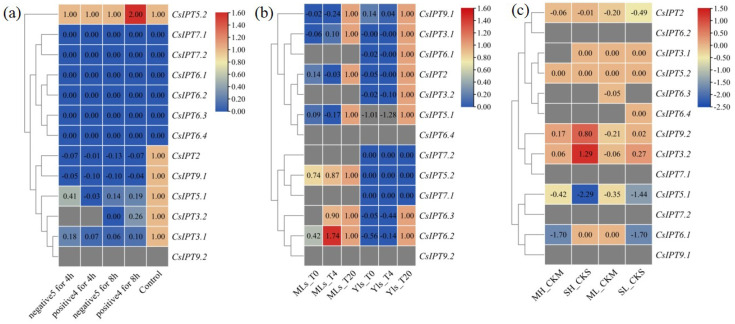
Expression profiles of *CsIPT* genes in tea plants in response to temperature stresses. Transcriptional levels of *CsIPT* genes under chilling and freezing stresses (**a**). Transcriptional levels of *CsIPT* genes under cold stresses (**b**). Transcriptional levels of *CsIPT* genes under high- and low-temperature stresses (**c**).

**Figure 9 plants-11-02243-f009:**
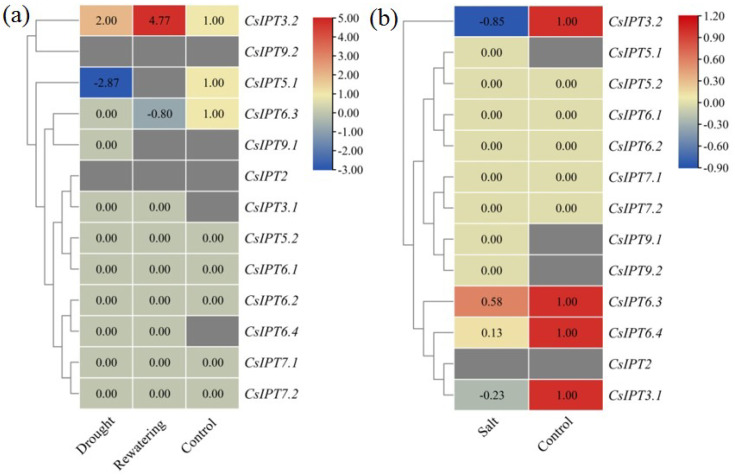
Expression profiles of *CsIPT* genes in tea plants in response to drought, rewatering, and salt stresses. Transcriptional levels of *CsIPT* genes under drought-stress and recovery (**a**). Transcriptional levels of *CsIPT* genes under salt-stress (**b**).

**Table 1 plants-11-02243-t001:** *IPT* gene family in *Camellia sinensis* L. LJ43 and SCZ, along with their molecular details and relevant genomic information.

Gene		CDS Length (bp)	Protein		
Name	Locus ID		Length (aa)	MW (kDa)	pI
CsIPT2	GWHPACFB004509 (CSS0044985.1)	1464		55.42	6.09
CsIPT3.1	GWHPACFB005294	858	285	32.68	8.81
CsIPT3.2	CSS0018660.1	852	283	31.76	7.10
CsIPT5.1	GWHPACFB026317 (CSS0005335.1)	1227	408	45.17	8.56
CsIPT5.2	GWHPACFB017705 (CSS0010471.1)	1062	353	39.66	5.77
CsIPT6.1	GWHPACFB021163	963	320	35.39	6.09
CsIPT6.2	CSS0006991.1	1104	367	41.69	8.99
CsIPT6.3	CSS0040585.1	1164	387	43.95	9.13
CsIPT6.4	GWHPACFB026680	447	149	15.90	6.21
CsIPT7.1	CSS0048745.1	909	302	34.31	6.08
CsIPT7.2	CSS0003027.1	555	184	20.63	5.11
CsIPT9.1	GWHPACFB024320	1383	400	45.73	6.25
CsIPT9.2	CSS0050211.1	2286	761	83.63	8.37

Abbreviation: MW = Molecular weight, pI = Isoelectric point.

## Data Availability

The data are available in the graphs and tables provided in the manuscript, in the Appendix A and tables, as well as downloaded from the NCBI Short-Read Archive (SRA) database (https://blast.ncbi.nlm.nih.gov/Blast.cgi accessed on 16 October 2020).

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
