# Peer review of "Genome-Wide Identification and Expression Analysis of Isopentenyl transferase Family Genes during Development and Resistance to Abiotic Stresses in Tea Plant (Camellia sinensis)"

_plants, 2022, doi:10.3390/plants11172243_

Round 1

Reviewer 1 Report

Cytokinin (CK) plays key roles in plant development and abiotic stress, IPT is a key enzyme of CK signaling. In the present work, IPT gene family was systematically investigated from tea plant genome database. The bioinformatics analysis was performed and their expression patterns were analyzed. The data could broadly support their conclusions. This study could provide a reference for further studies and contribute to tea cultivation management and molecular breeding. However, there are still some opinions/problems and misused typos that need to be solved and improved. I therefore would recommend a careful text check.

1.       The manuscript's abstract needs to be changed because it is quite unclear. The conclusion in the current investigation is ambiguous in the abstract. It should be written out that there are two classes of IPT genes and the gene numbers in each class. On the other hand, it should be written out the names of IPT genes which play key roles in the development and abiotic stress of tea plants, respectively.

2.       In Introduction, to better introduce the roles and research progress of IPT genes, I suggest that the first and second paragraph can be merged into one paragraph, and the sequence and logic of research progress and roles of IPT can be adjusted.

3.       Line 54-57, the research results of the CsADP/ATP‑IPT5 in the author’s recent study [13] should be clarified.

4.       Line 60-69, the introduction should be extended to discuss the hypothesis and research questions in detail.

5.       In ‘2. Materials and Methods’ section, the description of the method of paraffin section should be supplemented.

6.       In line 477-478, ‘CKs synthesized locally’ should be ‘locally synthesized CKs’.

7.       In line 481, ‘were’ should be ‘was’.

8.       In ‘3. Results and Discussion’ section, lines 709-722, the paragraph needs to be more concise.

9.       The conclusions need more explanation and should include the most significant findings, significance, implications and future work direction only.

Author Response

Many thanks for your critical comments and very valuable suggestions. We have worked through the details in the comments and revised the paper accordingly. The point-by-point responses are shown in the word.

Reviewer 2 Report

This article presented Genome-wide Identification and Expression Analysis of Isopentenyl Transferase Family Genes During Development and Resistance to Abiotic Stresses in Tea Plant (Camellia sinensis). The analyses recommended that IPT genes are involved in stress management in plants. This study will facilitate crop improvement in future. Before recommending this article for publication, there are some shortcomings for that should be resolve.

General comments

Overall, the study is well designed and presented in a good way, but mostly the literature is not cited.  

Abstract

The abstract is very short. The authors should highlight main findings in specific way and quantitative terms. Also add main methods used in this study.

Introduction

Line 37 add references.

Add role of CKs and IPT genes in other plant as well.

Also add its mechanism in pathways during stress.

Line 53-54 must be cited with the following studies.

https://doi.org/10.1016/j.plaphy.2021.01.042, https://doi.org/10.1007/s10725-021-00785-7,

The line must be cited with respective study “In our recent study, the full‑length cDNA of CsADP/ATP‑IPT5 (i.e. CsA-IPT5) was cloned”

Economic and medicinal importance of tea plant must be added. Also add stress responsive genes in tea.

Add mechanism of action of the respective genes in tea plant during stress.

Elaborate the role of IPT genes in stress management and transportation in specific tea plants.

Materials and Methods

Section 2.2 must be cited

Write full form of abbreviations at first use.

Results

Phylogenetic analysis should provide the details of clades.  

Overall, results are well presented.

Discussion

Discussion is well presented but some analyses are weekly compared with other studies. Like intron/exon pattern looks like results are written.

Conclusion

Conclusion is well justified.

Author Response

(The authors gave the same response as above.)

Reviewer 3 Report

Dear Editor,

I recommend a few corrections to the authors in the manuscript, which are listed below.

Chapter 2.6.2.  

Line: 145, 149

It would be worth mentioning, in case of low- and high-temperature stress, that values represent the day/night temperature.

Line 156: Authors write:

„The tea plants were subjected to severe drought stress and then allowed to recover after rewatering in field conditions, and the two and a bud were collected.“

In the second part of the sentence (underlined), there is no clear what part of the tea plants was collected except bud.

Line 159:

The expression "in vitro" write in italics.

Chapter 2.8

Data presented in the manuscript were statistically analysed. There is no information on how many plants or parallel samples were included in the statistical analysis.  

Chapter 3

Line 208 and 378:

The expression Arabidopsis write in italics as in the whole manuscript.

Line 205-208:

Authors compare the length and size of CsIPTs with IPTs in Arabidopsis and rice. There would be worth adding the reference to the article that includes this information.

I recommend checking the text to eliminate minor typing errors.

Author Response

(The authors gave the same response as above.)
